# Effect of Immature *Rubus occidentalis* on Postoperative Pain in a Rat Model

**DOI:** 10.3390/medicina59020264

**Published:** 2023-01-30

**Authors:** Geun Joo Choi, Hyun Kang, Oh Haeng Lee, Ji Wung Kwon

**Affiliations:** 1Department of Anesthesiology and Pain Medicine, Chung-Ang University College of Medicine, Seoul 06911, Republic of Korea; 2Gochang Berry & Bio Food Research Institute, Gochang 56417, Republic of Korea

**Keywords:** acute pain, black raspberry, multimodal analgesia, postoperative pain, *Rubus occidentalis*

## Abstract

*Background and Objectives*: This study aimed to identify the analgesic properties of immature *Rubus occidentalis* extract (iROE) using a postoperative-pain rat model. We also aimed to compare the analgesic effects of iROE to those of mature *R. occidentalis* extract (mROE) and examine the proinflammatory cytokine response and associated underlying mechanisms. *Materials and Methods*: In adult male Sprague Dawley rats, acute postoperative pain was induced through plantar hind-paw incisions. After the plantar incisions were made, the rats were intraperitoneally administered with normal saline or various doses of iROE and mROE to investigate and compare the analgesic effects of iROE and mROE. The mechanisms underlying iROE-induced analgesia were investigated via post-incisional administration of yohimbine, dexmedetomidine, prazosin, naloxone, atropine, or mecamylamine, followed by iROE. Mechanical withdrawal threshold (MWT) evaluations with von Frey filaments were carried out at different time points. Serum levels of tumor necrosis factor α, interleukin (IL)–1β, and IL-6 were measured to assess inflammatory responses. Multivariate analysis of variance (MANOVA) and linear mixed-effects model (LMEM) analysis were used to analyze the analgesic effect data. *Results*: The MWTs demonstrated significant increases in iROE in a dose-dependent manner up to 2 h after the plantar incisions were made. An LMEM analysis demonstrated that iROE yielded a significantly greater analgesic effect than mROE, but there was no significant difference between the two according to MANOVA. Dexmedetomidine enhanced the MWT-confirmed iROE response, while yohimbine and naloxone diminished it. Administration of iROE significantly attenuated the post-incisional increases in serum IL-1β and IL-6 levels. *Conclusions*: The iROE demonstrated analgesic and anti-inflammatory effects in a rat model of incisional pain, which were more pronounced than those associated with mROE. The analgesic activity of iROE may be associated with α_2_-adrenergic and opioid receptors.

## 1. Introduction

Postoperative pain control is a major issue in patient management following surgery and one of the most common pre-surgical patient concerns [1]. In a US national survey, up to 86% of adult patients who underwent surgery reported postoperative pain, with 75% reporting moderate or severe pain levels following surgery [2]. Postoperative pain can result in increased postoperative morbidity, delayed recovery, long-term opioid use, higher medical costs, and diminished quality of life [3]. Additionally, acute postoperative pain can become chronic if it is not properly managed [4]. Opioids have been a mainstay of postoperative pain management, but it can be challenging to maintain a balance between their usefulness in terms of pain control and their undesirable side effects [3,5]. As a result, there is an increasing need for pain management strategies that emphasize the use of multimodal approaches [5].

*Rubus occidentalis*, also known as black raspberry, has a variety of pharmacological properties—including anti-inflammatory and antioxidant properties—due to the bioactive compounds within it, such as anthocyanin and ellagic acid [6]. Phytochemical analysis has revealed that the concentrations of the fruit’s various constituent compounds vary with fruit maturity [7]. According to a study on different blackberry varieties, the number of phytochemical components and the degree of antioxidant activity decreased as the maturation stage progressed [8].

Our research team previously investigated the analgesic effects of mature *R. occidentalis* in a rat postoperative pain model [9]. However, to our knowledge, no published studies have investigated how immature *R. occidentalis* affects acute postoperative pain or how its effects vary with fruit maturity. Therefore, we hypothesized that immature *R. occidentalis* might be effective for pain control following surgery. We used a rat incisional pain model [10] to determine the immature *R. occidentalis’* analgesic effect and compare immature vs. mature *R. occidentalis* in terms of their analgesic effects. The primary outcome was to evaluate the analgesic effect of immature *R. occidentalis* in an experimental model of postoperative pain. The secondary outcomes were to investigate the difference in the analgesic effect between immature and mature *R. occidentalis,* the mechanisms underlying the analgesic activity of immature *R. occidentalis*, and its effect on inflammatory responses by measuring serum concentrations of proinflammatory cytokines.

## 2. Materials and Methods

This work was carried out in compliance with the National Institutes of Health Guide for the Care and Use of Laboratory Animals and was described in accordance with the Animal Research: Reporting In Vivo Experiments (ARRIVE) guidelines [11].

### 2.1. Preparation of the Immature and Mature R. occidentalis Extracts

The Gochang Black Raspberry Research Institute in South Korea provided the immature and mature *R. occidentalis* extracts (iROE and mROE, respectively). Dark-red mature fruits and green immature fruits were gathered after 38 post-bloom days and within 28 post-bloom days, respectively, from the Gochang (Jellabuk-Do) region of South Korea. Dr. Ji Wung Kwon, a research director at the Gochang Black Raspberry Research Institute where the voucher specimens of the two species were deposited, verified the authenticity of both immature and mature *R. occidentalis* specimens (the number of specimen voucher: GBRI-17, 18). 

The iROE and mROE were generated using the methods listed below [12]. We utilized 50% ethanol for fruit extraction since numerous studies on the *Rubus* species have shown it to be a reliable and efficient solvent for plant extraction. A reflux condenser was used to grind the immature and mature fruits, each weighing 1 kg, and extract them twice using 50% ethanol (10 weight for each fruit) at 80 °C for 2 h. The extract was concentrated and filtered before being lyophilized in a freeze-dryer and kept at 20 °C until use. For the immature and mature fruits, the extract yields were 17.2% and 7.5%, respectively. The iROE and mROE were produced just prior to intraperitoneal injection, and all operations were carried out in a sterile environment.

### 2.2. Study Animals

These investigations were carried out in Chung-Ang University’s Animal Research Laboratory and were authorized by the Institutional Animal Care and Use Committee (no. 2017-00102). Male Adult Sprague Dawley rats (250–300 g; Coretec, Seoul, Republic of Korea) were kept in individual cages in a temperature-controlled environment (22 °C) with 40% to 60% humidity and given access to tap water and a standard laboratory diet as needed. They were kept in 12 h light/dark cycles (with lights on from 8:00 a.m. to 8:00 p.m.) and given a week to acclimate to the housing facilities before the experiment. Females were excluded because hormonal changes may have altered their pain thresholds [13]. Rats with any kind of abnormality were excluded from the study. All rats were euthanized after the experiment via carbon dioxide inhalation [14].

### 2.3. Surgical Procedures

All surgical procedures were carried out under sterile conditions by single investigator who was uninformed of the group assignment. The rats were placed under general anesthesia with 6% isoflurane in 100% oxygen inside a sealed clear plastic chamber until they became motionless. Subsequently, the rats were kept on a nonrebreathing anesthetic circuit mask using 1–2% isoflurane in 100% oxygen. Before the incisions, Cefazolin (20 mg/kg; Chong Kun Dang Pharmaceutical Co., Seoul, Republic of Korea) was injected subcutaneously. Each rat’s plantar surface of the left hind paw was aseptically prepared for surgery. This study’s incisional pain model has been reported previously [10]. In brief, a blade was used to make a 1 cm longitudinal skin incision extending toward the digits on the plantar surface of the left hind paw, approximately 0.5 cm distal to the tibiotarsal joint. The plantaris muscle was separated, lifted slightly, and incised longitudinally. Two interrupted horizontal mattress sutures of 5-0 nylon were used to seal the wound.

### 2.4. Group Allocation and Blinding 

The rats were randomly divided into groups based on the tests carried out to assess iROE’s analgesic property and determine the mechanisms that underlie the iROE-induced analgesic effect. For three experiments, syringes carrying the study drugs were produced by an investigator not affiliated with the study in order to conceal the allocation. In 2 mL of normal saline, the study drugs were dissolved. For the control group, syringes with 2 mL of normal saline were generated. The syringes were wrapped in opaque tape and assigned sequential numbers based on a randomized list of related experiments. The prepared solutions were intraperitoneally injected in accordance with the study protocol. The investigators were blinded to the group assignment during all experimental procedures. 

### 2.5. Evaluation of iROE Analgesic Effect (Experiment 1)

Thirty rats were randomly assigned to five groups (*n* = six rats per group): the control group (with rats that received saline only) and groups with rats that received 10 mg/kg, 30 mg/kg, 100 mg/kg, and 300 mg/kg of iROE, respectively. Two hours after the plantar incisions, saline (0.9%) or various doses of iROE were administered intraperitoneally. The iROE doses were determined using logarithmic increments and were based on those reported in other experimental research on the pharmacological effects of *Rubus* species [9]. 

### 2.6. Elucidation of the Mechanism Underlying iROE-Induced Analgesia (Experiment 2)

The effects of iROE on mechanical hyperalgesia caused by plantar incision were further investigated to ascertain whether they were involved in α_1_ and α_2_ adrenergic, cholinergic (nicotinic and muscarinic), and opioid receptors. Forty-two rats were randomly allocated to seven groups (*n* = six rats per group), including one iROE group that served as a control and the other six groups that included rats that were given iROE and study drugs (yohimbine 2 mg/kg, dexmedetomidine 50 μg/kg, prazosin 1 mg/kg, atropine 5 mg/kg, mecamylamine 1 mg/kg, and naloxone 5 mg/kg). Two hours after the plantar incisions, normal saline or study drugs were administered intraperitoneally. After 10 min, 300 mg/kg iROE was administered intraperitoneally. Previous study suggested the application of drugs to investigate the possible involvement of the aforementioned receptor systems [9,15,16,17,18]. All study drugs were provided by Sigma-Aldrich.

### 2.7. Comparison of the Analgesic Effects of iROE and mROE (Experiment 3)

Twelve rats were randomly divided into two groups (*n* = six rats in each group), each receiving 300 mg/kg of either iROE or mROE. According to the findings of dose–response tests performed for this study as well as those for prior study, the iROE and mROE doses were set [9]. 

### 2.8. Comparisons of the Analgesic Effects of iROE and a Positive Control (Experiment 4)

The goal of this comparative experiment was to evaluate the study’s validity. Twelve rats were divided into two groups at random (*n* = six rats in each group). Ketorolac was applied as a reference analgesic to compare the iROE group to a positive control group. Ketorolac (30 mg/kg) was administered intraperitoneally 2 h after plantar incision [9,19]. The group that received the highest effective dose of iROE (300 mg/kg) for analgesia was compared to a group that received intraperitoneal ketorolac (30 mg/kg). 

### 2.9. Assessment of Motor Impairment (Experiment 5)

We applied an accelerated rotarod treadmill (Jeung Do Bio & Plant Co., Ltd., Seoul, Republic of Korea) to assess the motor and sedative effects of iROE. Twelve rats were allocated at random to one of two groups (*n* = six rats in each group): the iROE group and the control group. Two hours after the plantar incisions, the rats were given an intraperitoneal injection of 300 mg/kg of iROE or normal saline. The rotarod test was conducted 30 min, 60 min, 120 min, and 24 h after the injection of iROE or normal saline. The rats were placed on the rotarod, and the rotarod speed was gradually increased from 1 to 18 rotations per minute (rpm) over 120 s and then maintained at 18 rpm for another 30 s [9,20]. The time at which each rat fell off the rotarod was recorded.

### 2.10. Behavioral Measurements

Individual rats were placed on an elevated plastic mesh floor (with 8 × 8 mm perforations) beneath an upturned transparent plastic cage (21 × 27 × 15 cm) for 15 min to acclimate. Then, using von Frey filaments (Stoelting Co., Wood Dale, IL, USA), the rats’ withdrawal thresholds to mechanical stimulation were assessed. By applying pressure that caused the filament to gradually bend, the filaments were placed vertically with respect to the plantar surface of the hind paw. Filaments with bending forces of 4, 9, 20, 59, 78, 98, 147, and 254 mN were used until a rat retracted its hind paw or cutoff value of 254 mN was reached. Each filament was used three times at three-minute intervals. The mechanical withdrawal threshold (MWT) of the hind paw was measured by the lowest bending force that triggered paw withdrawal after the filament was applied [10]. A positive withdrawal response was defined as the full lifting of the plantar surface from the mesh floor. Following the observation of a response, filaments with varying bending forces were tested to confirm the MWT. Both behavioral testing and MWT assessment were performed by a single researcher who was not aware of the group assignment.

The following schedule was used to evaluate the MWTs: 1 day before the incision (baseline); 2 h after the plantar incision, i.e., immediately before iROE injection (after incision, AI); and 15 min, 30 min, 45 min, 60 min, 80 min, 100 min, 120 min, 24 h, 48 h, and 7 days after 0.9% saline or iROE injection.

### 2.11. Cytokine Assay

Eighteen rats were divided into six groups at random (*n* = thee rats in each group). Two hours after the plantar incision, the rats were injected intraperitoneally with 10 mg/kg iROE, 30 mg/kg iROE, 100 mg/kg iROE, 300 mg/kg iROE, 300 mg/kg mROE, or normal saline. Blood samples were collected from the lateral tail vein of the rats at 1, 2, 24, and 48 h following the injection of iROEs, mROE, or normal saline, and centrifuged at 2000× *g* for 15 min in a chilled, sterile EDTA tube (EDTA Vacutainer, Becton Dickinson, NJ, USA). The plasma was collected and kept at −80 °C until the tumor necrosis factor (TNF)-α, interleukin (IL)–1β, and IL-6 assays were performed. According to the manufacturer’s instructions, commercially available, enzyme-linked immunosorbent assay kits (R&D Systems, Minneapolis, MN, USA) were used to measure the serum level of TNF-α, IL-1β, and IL-6. 

### 2.12. Statistical Analysis

Sample size calculation was performed based on pilot study. The Shapiro–Wilk test was used to test for normality of variables. For abnormally distributed data, natural log transformation was performed, and natural log-transformed data passed the Shapiro–Wilk test. Therefore, we surmised that the normal distribution assumption for parametric test was not violated and decided to apply repeated measures ANOVA. Variables that passed Mauchly’s sphericity test were analyzed using repeated measures ANOVA, and Variables that did not pass Mauchly’s sphericity test were analyzed using one-way Wilk’s lambda multivariate analysis of variance (MANOVA). Individual measurements were expressed as the mean ± standard deviation and analyzed with SPSS 26.0 (IBM Corp., Armonk, NY, USA). A *p*-value of 0.05 or less was considered statistically significant. Detailed descriptions of statistical analyses are provided in Appendix A. The specifics were described in Appendix A. 

## 3. Results

### 3.1. Study Animals

The rats maintained good grooming behavior throughout the experiment and appeared to consume a typical amount of food and water. All of the rats completed the experiment without any complications. 

### 3.2. Evaluation of the Analgesic Effect of iROE

There were statistically significant differences between the groups, as shown by the Wilk’s lambda MANOVA results (F(48, 55.97) = 2.397, *p* < 0.001: Wilk’s lambda = 0.001, and partial η^2^ = 0.659) (Figure 1). Up to 48 h following surgery, the MWT displayed a growing concentration-dependent tendency in the iROE 10 mg/kg, iROE 30 mg/kg, iROE 100 mg/kg, and iROE 300 mg/kg groups.

In Figure 1, the MWT changes at baseline, AI, and 15 min, 30 min, 45 min, 60 min, 80 min, 100 min, 120 min, 24 h, 48 h, and 7 days following iROE administration are all depicted. When compared to the control, iROE 10 mg/kg, and iROE 30 mg/kg groups, the MWT values at 15, 30, 45, 60, 80, 100, and 120 min in the group administered iROE 300 mg/kg were significantly greater. Moreover, the MWT values in the 100 mg/kg group were significantly higher at 30 min than in the iROE 30 mg/kg group, and they were significantly higher at 120 min than in the control group. The MWT values at 15, 30, 45, 60, and 80 min for the iROE 30 mg/kg group were significantly greater than those in the control group, as were the MWT values at 15 and 60 min for the iROE 10 mg/kg group.

### 3.3. Elucidation of the Mechanism Mediating iROE-Induced Analgesia

The Wilk’s lambda MANOVA results revealed statistically significant differences among the groups (F(72, 136.379) = 3.003, *p* < 0.001: Wilk’s lambda = 0.006, and partial η^2^ = 0.577) (Figure 2a,b). Compared with the control group, the MWT values significantly decreased at 30, 45, 60, 80, and 100 min in the yohimbine group and at 15, 30, 45, 60, 80, 100, and 120 min in the naloxone group and increased at 80 min, 100 min, and 24 h in the dexmedetomidine group (Figure 2a,b).

The LMEM analysis showed statistically significant differences between the groups treated with prazosin, dexmedetomidine, yohimbine, naloxone, mecamylamine, and atropine (F(6, 412.03) = 69.864, *p* < 0.001) compared with group C (MD −0.23, 95% CI −0.36 to −0.10, *p* = 0.001; MD 0.45, 95% CI 0.32–0.58, *p* < 0.001; MD −0.67, 95% CI −0.80 to −0.54, *p* < 0.001; MD −0.69, 95% CI −0.82 to −0.56, *p* < 0.001; MD −0.17, 95% CI −0.30 to −0.04, *p* = 0.012; MD −0.24, 95% CI −0.37 to −0.11, *p* < 0.001, respectively).

### 3.4. Comparison of the Analgesic Effects of iROE and mROE

According to the MANOVA results, there was no evidence that the iROE 300 mg/kg group and mROE 300 mg/kg group differed significantly (F(10, 1) = 0.933, *p* = 0.675, Wilk’s lambda = 0.097, and partial η^2^ = 0.903). However, LMEM showed a statistically significant difference between the iROE and mROE groups. (F(1, 80.26) = 29.559, *p* < 0.001; MD 0.382, 95%CI 0.242–0.522) (Figure 3).

### 3.5. Comparison of the Analgesic Effects of iROE and the Positive Control

There was no evidence of a significant difference between the iROE 300 mg/kg group and the positive control group according to the MANOVA results (F(10, 1) = 1.915; *p* = 0.514, Wilk’s lambda = 0.050, and partial η^2^ = 0.950). LMEM showed no evidence of a difference between the iROE and positive control groups. (F(1, 63.404) = 3.753, *p* = 0.057; MD 0.178, 95% CI −0.006 to 0.365) (Figure 4).

### 3.6. Immunoassay

#### 3.6.1. IL-1β

Repeated-measures ANOVA showed significant differences with respect to IL-1β among the control group, the iROE 10 mg/kg group, the iROE 30 mg/kg group, the iROE 100 mg/kg group, the iROE 300 mg/kg, and the mROE group (F(5,12) = 6.609, *p* = 0.004) (Figure 5a). The serum levels of IL-1β in groups iROE 300 mg/kg and mROE 300 mg/kg were significantly attenuated compared to group C and the iROE 10 mg/kg group at 24 h after surgery.

The LMEM results revealed statistically significant differences among the groups (F(5,66) = 17.67, *p* < 0.001), specifically with respect to the iROE 30 mg/kg, iROE 100 mg/kg, iROE 300 mg/kg, and mROE 300 mg/kg groups (19.65 (2.05–37.25), *p* = 0.029: 21.84 (4.23–39.44), *p* = 0.016: 58.41 (40.81–76.01), *p* < 0.001: 41.31 (23.71–58.91), *p* < 0.001) compared with group C.

#### 3.6.2. IL-6

Repeated-measures ANOVA showed significant differences with respect to the IL-6 levels among the control group, iROE 10 mg/kg group, iROE 30 mg/kg group, iROE 100 mg/kg group, iROE 300 mg/kg, and mROE group. (F(5,12) = 11.625, *p* < 0.001). The serum level of IL-6 was significantly attenuated in the iROE 100 mg/kg, iROE 300 mg/kg, and mROE 300 mg/kg groups compared to group C at 24 and 48 h after surgery; in the iROE 100 mg/kg, iROE 300 mg/kg, and mROE 300 mg/kg groups at 24 h after surgery; and in the iROE 300 mg/kg group 48 h after surgery compared to group C (Figure 5b).

The LMEM results revealed statistically significant differences among the groups (F(5,66) = 5.808, *p* < 0.001), specifically for the iROE 100 mg/kg, iROE 300 mg/kg, and mROE 300 mg/kg groups (14.19 (0.70–26.25), *p* = 0.042: 25.67 (10.78–40.56), *p* = 0.001:15.77 (0.89–30.66), *p* = 0.038) compared with group C.

#### 3.6.3. TNF-α

There were no significant differences with respect to TNF-α levels among the control group, iROE 10 mg/kg group, iROE 30 mg/kg group, iROE 100 mg/kg group, iROE 300 mg/kg, and mROE group (F(5,12) = 1.042, *p* = 0.437). 

The LMEM results revealed statistically significant differences among the groups (F(5,66) = 2.901, *p* = 0.020), specifically for the iROE 300 mg/kg group (35.22 (3.53–66.91), *p* = 0.030) compared with group C (Figure 5c).

### 3.7. Assessment of Motor Impairment

Repeated-measures ANOVA and LMEM showed that there was no evidence of a difference in rotarod time between the control group and the iROE 300 mg/kg group (F(1,4) = 0.006, *p* = 0.941, and F(1,27.542) = 0.079, *p* = 0.781; MD 1.08, 95% CI −6.83 to 8.99) (Figure 6).

## 4. Discussion

Our findings showed that the intraperitoneal administration of iROE had a concentration-dependent analgesic effect in a rat model of incisional pain. Using the same experimental model as in our previous study, we confirmed the analgesic effect of mature *Rubus occidentalis* in a concentration-dependent manner [9]. Black raspberries contain a high concentration of bioactive phytochemicals such as anthocyanins, ellagic acid, and flavonoid compounds [21]. Based on these phytochemicals, numerous studies have shown that *R. occidentalis* has a variety of biological effects that may be beneficial to human health. Accordingly, anti-oxidative, anti-inflammatory, anti-nociceptive, and anti-cancer properties have been demonstrated [6,22]. 

In this study, the analgesic effect of immature *R. occidentalis* observed in an incisional pain rat model may provide insightful evidence in terms of postoperative pain management. Although surgery is an essential medical procedure that reduces the risk of death and disability, postoperative pain is unavoidable and occurs in varying degrees. Thus, postoperative pain control is a major concern, and current clinical practice guidelines recommend the induction of multimodal analgesia to reduce the adverse effects associated with the use of opioids and to achieve more effective pain control [2,3]. That is, multimodal analgesia is a pain-relief strategy regulating pain via multiple mechanisms based on different nociceptive pathways [23]. In this regard, evidence for various analgesic candidates applicable to multimodal analgesia is obviously required. Furthermore, the identification of analgesic candidates based on food-based bioactive species is expected to broaden the scope and efficacy of multimodal analgesia strategies [24]. Fruits, vegetables, and other plant foods contain phytochemicals, which are nutrient-rich bioactive plant chemicals that may confer beneficial effects on human health. Black raspberry contains great quantities of polyphenols, flavonoids, and anthocyanins, which are among the phytochemicals that may provide health benefits [25].

Given that these properties can vary depending on the fruit’s maturity state, in the present study, we endeavored to determine whether the immature fruit had an analgesic effect, and if so, to compare the immature fruit with the mature fruit. As a result, iROE indicated a superior potential for conferring an analgesic effect on mROE in a maturity-related comparison. Although MANOVA did not reveal a statistically significant difference, the LMEM showed such a difference between the analgesic effect of iROE and mROE. We are unable to conclusively suggest that iROE has a significantly greater analgesic effect than mROE because MANOVA was the primary method of statistical analysisand LMEM wasa secondary analysis method. However, it is necessary to discuss the implications of the significant findings from the LMEM, which can be more strongly supported by the findings of several other studies on the maturity of *Rubus* fruits [8,26,27,28]. 

The concentration of phenolic compounds varied with fruit maturity according to a study that involved the phytochemical extraction and analysis of the phenolic profile of the fruit [29,30]. According to one study that used different blackberry cultivars, maturity was used to predict the possible content of secondary metabolites with antioxidant capacity. Furthermore, as the maturity stage progressed, blackberries’ phytochemical component content and antioxidant activity decreased [8]. Another study found that immature *Rubus* species had higher antioxidant properties than mature fruits, which was attributed to the decomposition of antioxidant-related compounds during ripening [31]. According to the findings of a study on the relationship between antioxidative effects and postoperative pain, antioxidants reduced pain and the requirement for analgesia after foot and ankle trauma surgery [32] and laparoscopic colectomy [33]. These findings are consistent with those of the present study, which show the greater potential in terms of analgesic effects of iROE compared to mROE. 

This is also related to our experiment on the inflammatory response. In this study, iROE significantly reduced the levels of proinflammatory cytokines at 24 and 48 h after surgery, and iROE demonstrated a stronger anti-inflammatory potential than mROE. Oxygen-derived free radicals cause toxic oxidation reactions in cells, which can cause inflammatory damage [34]. As a result, the greater antioxidant effect of iROE mentioned above is linked to its greater anti-inflammatory effect. According to one study, immature *Rubus* fruit contained more ellagic acid than the mature fruit, indicating that the ellagic acid component of the immature fruit inhibited the production of proinflammatory cytokines and inflammation-related factors in cells [35]. An incision made during surgery damages tissue, causes inflammation, and produces pain. Consequently, acute postoperative pain is a natural physiological reaction to tissue damage, and inflammation is both one of the most common causes of postoperative pain and one of its most potent potential triggers. The degree of acute pain following surgery will increase if the corresponding inflammation is not adequately controlled, and recovery will be delayed [36]. Furthermore, this can be associated with chronic pain, and patients may suffer from pain and its related complications for a long time [3,4]. Thus, it is anticipated that the anti-inflammatory and analgesic properties of iROE presented in our research will be beneficial candidates for postoperative pain management. Furthermore, a comparison using ketorolac, a representative analgesic with an anti-inflammatory effect [37], as a positive control showed no significant difference between the analgesic effects of ketorolac and iROE, demonstrating the benefit of iROE in postoperative pain management. 

Multimodal analgesia represents a more systematic approach to postoperative pain management that is based on the analgesic mechanisms of potential candidates [5]. We also investigated the mechanisms of iROE’s analgesic effect, which was enhanced by dexmedetomidine and antagonized by yohimbine and naloxone. This suggests that the analgesic activity of iROE may be mediated by the α_2_-adrenoreceptor and the opioid receptor. Multimodal analgesia can be performed based on this analgesic mechanism, taking into account additive and/or synergistic effects with respect to attenuating pain pathways [23].

This study has some limitations. First, the incisional pain rat model used in this study cannot represent all types of surgery. Preclinical studies involving animal models capable of performing abdominal or pelvic surgery involving visceral pain as well as parietal pain will be required. Second, we did not perform an experiment at the molecular or cellular level to explain the antioxidant and antinociceptive activities of iROE. Since our study was the first to suggest the effect of iROE in a postoperative pain model, the evidence from this study can lead the way for future research regarding the analgesic properties of black raspberry. Third, it may be difficult to fully support the potential of the clinical applicability of black raspberry based solely on the intraperitoneal administration of iROE. The intraperitoneal route, on the other hand, is simple to establish, conducive to rapid procedures, and its use induces a less stressful effect on laboratory rodents, making it useful for pain studies [38]. Given current knowledge, it is expected that a preclinical study investigating the effect of iROE via the oral route will bolster the current evidence in terms of clinical applicability. 

Nonetheless, this study has strengths in that it is the first study to confirm the effect of immature *R. occidentalis* in a postoperative pain model based on a previously established and rigorously designed study protocol. Various studies on the application of black raspberry for postoperative pain management can be suggested based on the evidence presented in this study. Furthermore, this research is expected to benefit human health in the clinical field. 

## 5. Conclusions

Immature *R. occidentalis* demonstrated analgesic and anti-inflammatory effects in a rat model of incisional pain, indicating a superior potential compared to mature *R. occidentalis*. This suggests that immature *R. occidentalis* could be a promising candidate for the development of multimodal analgesia strategies for postoperative pain management. The analgesia induced by immature *R. occidentalis* may be associated with α_2_-adrenergic and opioid receptors.

## Figures and Tables

**Figure 1 medicina-59-00264-f001:**
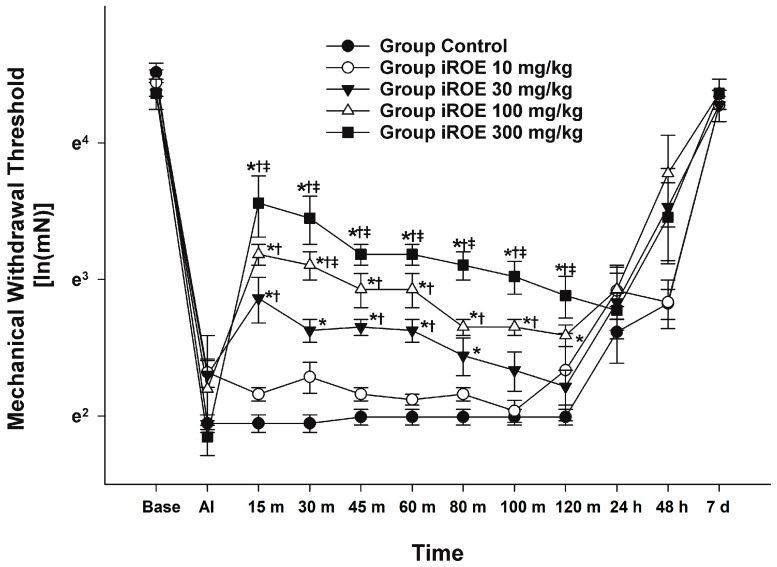
Analgesic effects of immature *R. occidentalis*. * *p* < 0.05 compared with the control group, ^†^
*p* < 0.05 compared with iROE 10 mg/kg group, and ^‡^
*p* < 0.05 compared with iROE 30 mg/kg group. iROE, immature *Rubus occidentalis* extract; AI, after incision.

**Figure 2 medicina-59-00264-f002:**
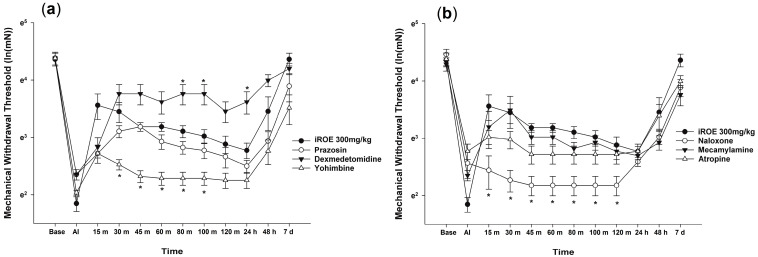
(**a**) Analgesic mechanism of iROE together with yohimbine, prazosin, or dexmedetomidine and (**b**) analgesic mechanism of iROE together with atropine, mecamylamine, or naloxone, * *p* < 0.05, compared with the iROE 300 mg/kg group. iROE, immature *Rubus occidentalis* extract; AI, after incision.

**Figure 3 medicina-59-00264-f003:**
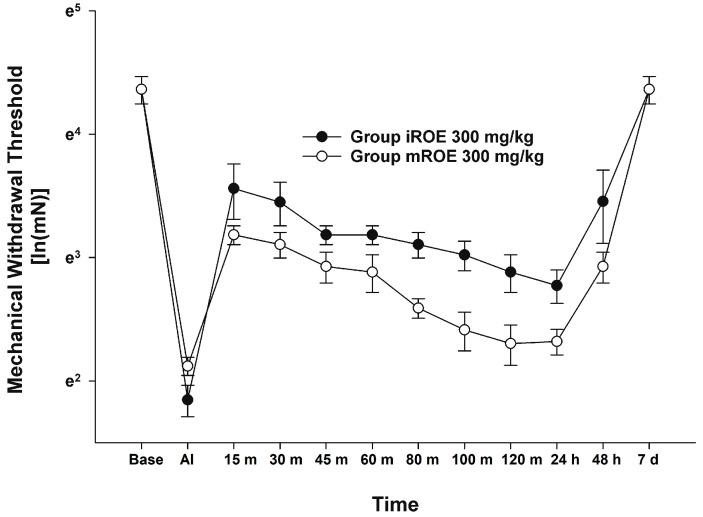
Analgesic effect of immature *R. occidentalis* versus mature *R. occidentalis.* iROE, immature *Rubus occidentalis* extract; mROE, mature *Rubus occidentalis* extract; AI, after incision.

**Figure 4 medicina-59-00264-f004:**
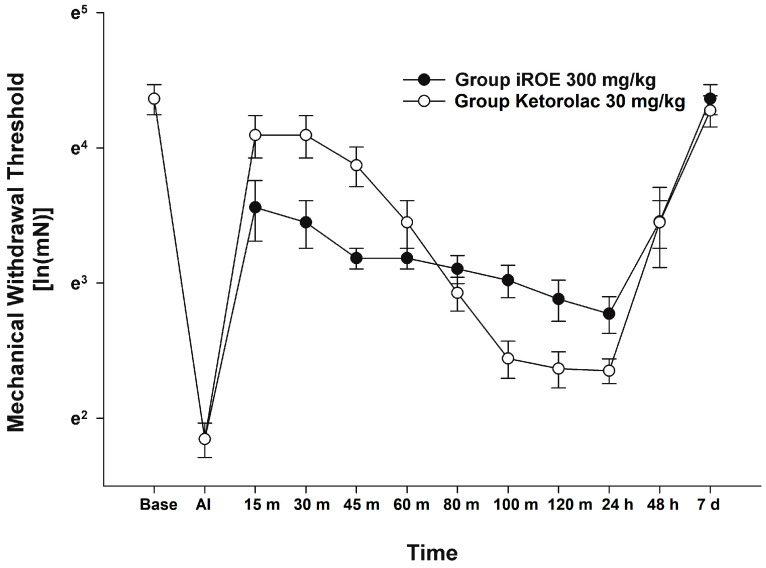
Analgesic effect of immature *R. occidentalis* vs. ketorolac. iROE, immature *Rubus occidentalis*; AI, after incision.

**Figure 5 medicina-59-00264-f005:**
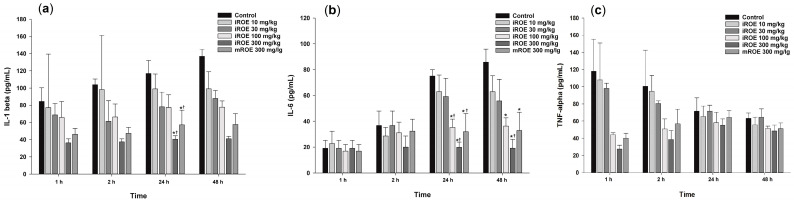
(**a**) Effect of immature *R. occidentalis* on IL-β, (**b**) effect of immature *R. occidentalis* on IL-6, and (**c**) effect of immature R. occidentalis on TNF-α; * *p* < 0.05 compared with the control group and ^†^
*p* < 0.05 compared with iROE 10 mg/kg group. IL, interleukin; TNF, tumor necrosis factor; iROE, immature *Rubus occidentalis* extract; mROE, mature *Rubus occidentalis* extract.

**Figure 6 medicina-59-00264-f006:**
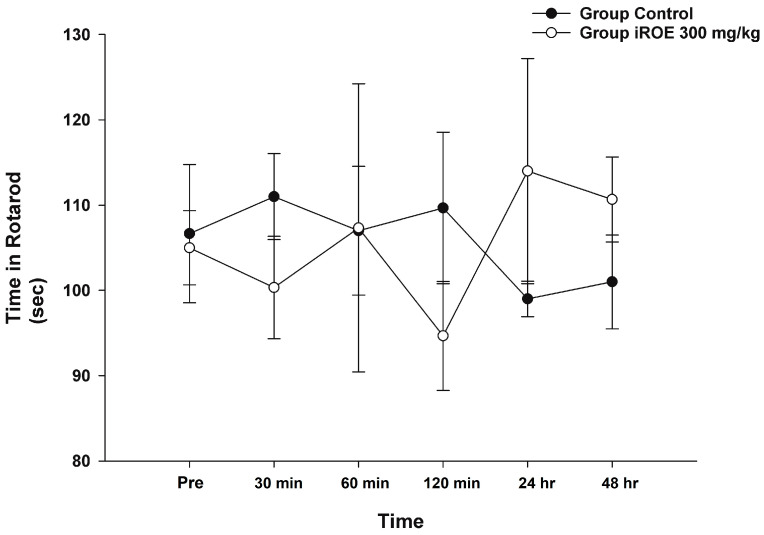
Effect of immature *R. occidentalis* on Rotarod testing. iROE, immature *Rubus occidentalis* extract.

## Data Availability

The data of this study are available from the corresponding author upon reasonable request.

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
