# Peer review of "Effect of Immature Rubus occidentalis on Postoperative Pain in a Rat Model"

_medicina, 2023, doi:10.3390/medicina59020264_

Round 1

Reviewer 1 Report

This study is well made and it has provided good options for future possible multimodal pain therapy. Aim of the study (primary and secondary) are well pointed. The methods and metodology are well described and followed by recent references. I congratulate the authors for well made paper. 

As you pointed out the rats hindpaw is very well inervated but it may not adequately provide the neccesary end point for all of the procedures so I urge you to make a different types (models) for pain assesment which is well needed to provide better results.

Maybe for future perspetive it can be used in human models to see it applicability - it is well defined that rat models sometimes are very different from human model studies.

Author Response

Thank you for your constructive and considerate review of our work. According to your comment, we revised our manuscript for the different types of animal model for pain assessment in the future research. [Line 395 to 398 on page 10 to 11]

Reviewer 2 Report

This article evaluated the effects of immature Rubus occidentalis on postoperative pain in a rat model. There are some issues in this manuscript that should be addressed as follows:

·         Abstract:

1. The subheadings “Background and Objectives” should be replaced with “Objectives”.

2. Page 1 Line 10: The word “rate” should be replaced with “rat”.

3. The results regarding serum tumor necrosis factor-α should be mentioned in the abstract.                                     

·         Introduction:

1.    The novel points in this study should be explained in the introduction section because there are previous studies by the same authors that discussed a similar issue; e.g. https://pubmed.ncbi.nlm.nih.gov/27916366/, https://www.ncbi.nlm.nih.gov/pmc/articles/PMC4940828/

2.    The aim of the study mentioned in the “Introduction” section should be similar to that mentioned in the “Abstract”.

·         Materials and methods:

1.    The source and CAT number of the used kits, drugs, and chemicals should be mentioned.

2.    Page 2: A reference should be added to the method of preparation of iROE and mROE..

3.    How did you know that the animals were acclimatized?

4.    The degree of humidity at which the animals were housed should be mentioned.

5.    Page 4: A reference should be added to the behavioral measurements.

6.    Statistical analysis should be summarized to be easily understood.

·         Results: A collective diagram summarizing the main findings of this study is recommended.

·         Conclusion: The possible implication of the findings of the present study in the clinical settings should be mentioned.

·         General comments:   

1. The manuscript should be revised by English-naïve speaker to improve the quality of the language.

2. The manuscript should be checked regarding the grammatical errors and plagiarism.

Author Response

This article evaluated the effects of immature Rubus occidentalis on postoperative pain in a rat model. There are some issues in this manuscript that should be addressed as follows:

Abstract:

1. The subheadings “Background and Objectives” should be replaced with “Objectives”.

Our response] We revised it according to your comment.

2. Page 1 Line 10: The word “rate” should be replaced with “rat”.

Our response] Thank you for your pointed comment. We corrected it.

3. The results regarding serum tumor necrosis factor-α should be mentioned in the abstract. 

Our response] As the result of TNF-α did not show a significant result and the limitation of the word number of abstract, we did not include the results of TNF- α. We tried to modify the result considering your comment. [Line 26 on page 1]

Introduction:

1. The novel points in this study should be explained in the introduction section because there are previous studies by the same authors that discussed a similar issue; e.g. https://pubmed.ncbi.nlm.nih.gov/27916366/, https://www.ncbi.nlm.nih.gov/pmc/articles/PMC4940828/

Our response] There has been no evidence regarding the effect of immature R. occidentalis on postoperative pain model. Hence, we purposed to provide the evidence in addition to our previous findings in this study [Line 54 to 56 on page 2]

2. The aim of the study mentioned in the “Introduction” section should be similar to that mentioned in the “Abstract”.

Our response] We specified the aim of this study in the introduction section according to your comment. [Line 59 to 64 on page 2]

Materials and methods:

1. The source and CAT number of the used kits, drugs, and chemicals should be mentioned.

Our response] We revised the manuscript according to your comment. [Methods section]

2. Page 2: A reference should be added to the method of preparation of iROE and mROE.

Our response] The reference was added.

3. How did you know that the animals were acclimatized?

Our response] Acclimation period was 7 days, as measured by the physiological parameters of body weight and activity. This was based on our institute protocol.

4. The degree of humidity at which the animals were housed should be mentioned.

Our response] Humidity was maintained between 40% to 60%, which was added in the manuscript. [line 91 on page 2]

5.  A reference should be added to the behavioral measurements.

Our response] The reference was added.

6. Statistical analysis should be summarized to be easily understood.

Our response] We summarized the statistical analysis, and described the specifics in the supplementary file S2. 

Results: A collective diagram summarizing the main findings of this study is recommended.

Our response] Thank you for your considerate comment for the betterment of our manuscript. We added a collective diagram using AUC (area under the curve regarding mechanical withdrawal threshold of immature and mature ROE, was showed in supplementary file S3.

Conclusion: The possible implication of the findings of the present study in the clinical settings should be mentioned.

Our response] Thank you for your constructive comment. We described the potential implication in the clinical setting. [line 454 to 455 on page 12]

General comments:   

1. The manuscript should be revised by English-naïve speaker to improve the quality of the language.

Our response] We got the English editing service by native speaker additionally. The certificate was attached as a reference. 

2. The manuscript should be checked regarding the grammatical errors and plagiarism.

Our response] We identified through plagiarism check service once more.   
